# Investigations for a *Yarrowia*-Based Biorefinery: In Vitro Proof-of-Concept for Manufacturing Sweetener, Cosmetic Ingredient, and Bioemulsifier

**Edina Eszterbauer and Áron Németh \*** 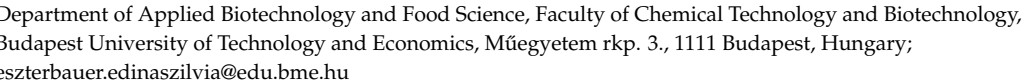

Department of Applied Biotechnology and Food Science, Faculty of Chemical Technology and Biotechnology, Budapest University of Technology and Economics, Műegyetem rkp. 3., 1111 Budapest, Hungary; eszterbauer.edinaszilvia@edu.bme.hu

\* Correspondence: naron@f-labor.mkt.bme.hu; Tel.: +36-1-463-2595

**Abstract:** *Yarrowia lipolytica* is a widely used microorganism in biotechnology since it is capable of producing a wide range of products (lipase, citric acid, polyols). A less-studied related strain is *Y. divulgata*, which is also capable of erythritol production in even higher concentration than most *Y. lipolytica* wild strains from glycerol as renewable feedstock. Thus, the aim of this work was to investigate *Y. divulgata's* complex utilisation based on erythritol fermentation from glycerol to establish a *Yarrowia*-based biorefinery in which both the fermentation broth and separated cells are converted into high added-value products (erythritol, bioemulsifier, cosmetic ingredient, i.e., skin moisturizer). An important parameter of erythritol fermentation is an adequate oxygen level, so both the constant oxygen level and oxygen absorption rate were investigated regarding the three target products. DO (dissolved oxygen) = 10, 20, 30, 40% was examined in the bioreactor, and a KLa range of 18–655 h$^{-1}$ was investigated in both the bioreactor and in different types of shaking flasks, applying two different glycerol levels (100–150 g/L). The results showed that the *Yarrowia divulagata* NCAIM 1485 strain could produce one of the highest amounts of erythritol (44.14 $\pm$ 1 g/L) among wild-type yeasts from 150 g/L glycerol beside a $K_L$a value of 655 h$^{-1}$. Cell-lysates skin hydrating activity was the highest (12%) when DO = 20% ($K_L$a 26.4 h$^{-1}$) was applied. In all cases, the collected samples had an emulsification index above 69% which did not decrease below 54% after 24 h, showing good stability. Since *Y. divulgata* fermentations resulted in three high added-value products at the same time from a renewable raw material (glycerol), we concluded that it is suitable for complex utilisation in a microbial biorefinery, since the fermentation broth can be used for the isolation of a sweetener and bioemulsifier; meanwhile, the separated cells can be processed for cosmetic application as a skin moisturizer.

**Keywords:** *Yarrowia divulgata*; fermentation; oxygen transfer; erythritol; cell-lysate; biodetergent; glycerol; skin moisturizer

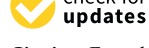



## 1. Introduction

Biorefineries are defined as the sustainable processing of biomass into a spectrum of marketable products and energy [1]. *Yarrowia lipolytica* is a good candidate since it is used widely in biotechnology [2], utilizing its wide degradation capabilities and product spectrum. The first use was in 1960s, when it was used to produce single-cell proteins (SCP) using n-alkanes as a carbon source [3,4]. *Y. lipolytica* also has strong proteolytic and lipolytic activity as well [5,6]. It is capable of producing and secreting a number of metabolites that can be used in different industries. *Y. lipolytica* has attracted interest because of its ability to produce metabolites of high commercial value, such as organic acids (e.g., citric acid, isocitric acid, α-ketoglutaric acid, succinic acid), single-cell oil and aromatic compounds (e.g., lactones and esters), enzymes (e.g., protease, phosphatase, lipase, esterase), and heterologous proteins (e.g., laccase, epoxide hydrolase) [7]. For example, *Y. lipolytica* is

used to produce nutrient biomass as it is an extensive source of protein and is a safe food supplement even for humans as it has GRAS status. Jach and Malm observed in their research that increasing aeration increased the activity of acyl-CoA oxidases, so the carbon flux was used for acetyl-CoA synthesis and then for the growth of defatted protein biomass. Thus, culture conditions (e.g., agitation and aeration) were found to contribute to the production of protein-rich biomass [8].

Also, several studies have shown that *Yarrowia lipolytica* is able to produce erythritol together with mannitol and citric acid by fermentation [9–11].

The growth of *Y. lipolytica* and the quantity, as well as the quality of the different metabolites it secretes, are influenced by various environmental factors. The environmental conditions, namely physicochemical (pH, temperature, dissolved oxygen concentration and, osmotic stress), mechanical (agitation rate and pressure), and nutritional effectors (carbon and nitrogen sources and metal ions), have an impact on morphology and product formation (enzymes, citric acid, and lipid) [12]. *Y. lipolytica* is a strictly aerobic microorganism. One of the most important factors is the amount of oxygen, which plays an important role in the growth of the microorganism, so aeration is essential for high growth rates [13].

*Yarrowia divulgata* is a recently described species which can be isolated from meat of animal origins, among other sources, but is also found in oceanic fish. *Y. divulgata* NCAIM 1485 was isolated from chicken liver in 1999 and described as a strain that is phenotypically indistinguishable from *Y. lipolytica* and *Y. deformans*, the latter being the closest genetic relative. A few studies have begun to investigate the ability of the strain to produce erythritol using a glycerol carbon source [14–16].

Erythritol ($C_4H_{10}O_4$) is a naturally occurring four-carbon sugar alcohol, also known as a polyol, which is widely used in the food industry. It occurs naturally in small amounts in alcoholic beverages, mushrooms, and fruits such as pears, grapes, and watermelon. However, sugar alcohols and functional sugars are present in smaller amounts in nature (in plants, fungi, and algae). In plants, sugar alcohols accumulate temporarily in leaves during light hours and are transported to other organs during darkness. The low content makes the extraction of sugar alcohols from plants difficult [17]. Erythritol is 60–70% as sweet as table sucrose yet it has a very low caloric value (0.2 kcal/g), has no effect on human blood sugar levels, and does not cause tooth decay [18]. Erythritol is well absorbed and does not ferment, so it can be consumed in relatively large quantities without side effects, unlike other polyols, which can cause gastrointestinal side effects if consumed in large quantities [19,20]. Erythritol first appeared on the Japanese market in 1990 and has since been approved for food use in more than 60 countries. Within the food industry, erythritol is mainly used as a sweetener in finished products, improving organoleptic characteristics such as taste, colour, and texture. Additionally, as a sugar substitute, erythritol can be found in tabletop sweeteners, beverages, chewing gums, chocolates, candies, and bakery products, resulting in reduced calorie-containing products. Due to its properties, erythritol, as an excipient, provides good fluidity and stability. Therefore, it is increasingly used as excipients incorporated into personal care products such as toothpaste, mouthwashes, creams, make-up, perfumes, and deodorants. Finally, erythritol is also widely used in the pharmaceutical industry both in solid and liquid formulations, including granulated powders, tablets, tablet extracts, medicinal chewing gums, syrups, and the oral care products mentioned above [21]. The global erythritol market size exceeded USD 195 million in 2019 and is estimated to reach USD 310 million by 2026 [22]. The current world production of erythritol is around 60,000 t/year [23].

Many methods are available for the production of erythritol, both chemical and biotechnological. One of the known chemical production methods, which produces equimolar amounts of erythritol and ethylene glycol, is a multi-step process involving chemical synthesis from dialdehyde starch at a high temperature in the presence of a metal catalyst, namely nickel. However, it is not used industrially due to its very low yield and relatively high costs [18].

Erythritol for industrial use is produced by microbial methods using osmophilic yeasts (*Moniliella pollinis*, *Trichosporonoides megachiliensis*, and *Y. lipolytica*) and some bacteria (*Oenococcus oeni*, *Leuconostoc mesenteroides* and *Lactobacillus sanfranciscensis*) [24]. Table 1 shows the results obtained in the production of erythritol by different *Yarrowia* strains.

**Table 1.** Erythritol fermentation with wild-type and mutant *Yarrowia* strains.

| Wild-Type Strains | Carbon Source | Cultivation Strategy | Titer | References |
|---|---|---|---|---|
| *Y. lipolytica* A-15 | 100 g/L glycerol | Batch bioreactor | 28 g/L | [25] |
| *Y. lipolytica* A-3 | 150 g/L glycerol | Batch bioreactor | 25.3 g/L | [26] |
| *Y. lipolytica* A-6 | 150 g/L glycerol | Batch bioreactor | 32 g/L | [26] |
| *Y. divulgata* CBS11013 | 100 g/L glycerol | Batch bioreactor | 35.4 g/L | [15] |
| **Mutant Strains** | **Carbon Source** | **Cultivation Strategy** | **Titer** | **References** |
| *Y. lipolytica* MK1 UV mutant | molasses and glycerol | Two-stage fermentation | 113.1 g/L | [27] |
| *Y. lipolytica* Wratislavia K1 | glycerol | Fed-batch | 81 g/L | [28] |
| *Y. lipolytica* CICC 1675 | glycerol | One-stage fed-batch fermentation | 194.3 g/L | [10] |

Most of the high erythritol production has been achieved by mutant strains, and metabolic engineering also represents an important step forward to enhance erythritol production. Yang et al. [29] produced high erythritol levels (150 g/L) by overexpressing glycerol kinase (*GUT1*), glycerol-3-P dehydrogenase (*GUT2*), and transketolase (*TKL1*) while knocking out erythritol dehydrogenase (*EYD1*), which is in the erythritol catabolic pathway. Janek et al. [30] investigated the effect of erythrose reductase enzyme overexpression on erythritol production. This enzyme reduced erythrose to erythritol by NAD(P)H oxidation. The strain (wild-type *Y.lipolytica* A101) was transformed with the *YALI0F18590g* gene, which encoded and overexpressed the erythrose reductase enzyme, thus producing 44.4 g/L erythritol compared to the control (MK1), which reached 37.1 g/L. Further studies were carried out to investigate the effect of zinc on erythritol production and enzyme function, and researchers found that the addition of zinc further increased erythritol production in both the mutant and control (MK1) strains, resulting in 54.1 g/L and 51 g/L, respectively.

Osmotolerant yeasts are able to survive osmotic stress by accumulating various solutes of high osmolarity, called osmolytes. These substances protect and stabilize their enzymes and allow essential cellular functions to operate properly under higher osmotic pressure. The most common osmolyte is glycerol in yeasts, but sugar alcohols such as erythritol and mannitol can also serve as osmolytes [31]. Da Silva et al. [32] looked for an alternative supplement to NaCl to increase osmolarity. They found that polyethylene glycol (PEG) is a good osmotic agent. They used a concentration to achieve the same osmolarity as with 25 g/L NaCl supplementation. The application of PEG resulted in an increase in erythritol productivity: *Y. lipolytica* W29 produced 42.5 g/L (44% increase), and *Y. lipolytica* IMUFRJ 50,682 produced 25.38 g/L (40% increase) of erythritol. Jagtap et al. [33] showed that supplementing the glycerol carbon source with NaCl increased the amount of erythritol and led to perfect utilization of glycerol, as opposed to glucose. Quantitative PCR analysis clearly demonstrated that the expression of genes involved in the glycerol uptake and utilization mechanism (e.g., glycerol kinase (*GK*), glycerol dehydrogenase (*GCY1*)) was increased in the presence of salt. The combined overexpression of sugar alcohol phosphatase (*PYP*), glycerol kinase (*GK*), and transketolase (*TKL*) further increased the glycerol utilization rate and erythritol titre. In the case of *Yarrowia lipolytica* PO1f-*PYP-GK-TKL* strain, 58.8 g/L erythritol was achieved, which was a 1.9-fold increase in erythritol compared to the wild-type strain (*Y. l.* PO1f).

Ribeiro [34] studied the effect of aeration on erythritol production. After selection, he found that strain *Y. lipolytica* W29 produced the most erythritol (34 g/L) from 100 g/L glycerol at 200 rpm at 27 °C. Among the strains tested, *Y. divulgata* 5257/2 produced 17 g/L erythritol under the same conditions. The test strain (*Y. lipolytica* W29) produced 35 g/L erythritol in 72 h at 3 vvm and 900 rpm aeration and agitation rate and consumed all glycerol from the medium, with a productivity of 0.5 g/L/h. Machado [35] investigated the erythritol production of wild-type strain *Y. lipolytica* W29 at dissolved oxygen concentrations (20% and 40%), and found that at 20% oxygen, 21.33 g/L erythritol was produced by hour 55 but was later consumed by the strain up to the end of fermentation. At a 40% oxygen level, the strain produced 23.3 g/L of erythritol. Upon further investigation, the strain produced 31.8 g/L erythritol at 900 rpm and 3 vvm aeration. In a fed-batch fermentation (900 rpm and 3 vvm aeration) at the best setting, 55.4 g/L of erythritol was produced in nearly 200 h. It has been shown that the concentration of dissolved oxygen affects the amount of erythritol in the fermented media, so a higher percentage of dissolved oxygen increases erythritol production.

*Yarrowia lipolytica* is also capable of producing products that can be used in the cosmetic industry. This includes, among others, aroma components such as limonene, which is widely used in the cosmetic industry [36]. It is also capable of producing pigments, which are also used in the cosmetic industry, for example, as skin tanning products. The produced pigment is pyomelanin, which is a brown pigment, so it can even be used in sunscreen creams [37]. The *Yarrowia lipolytica* fermentation lysate, which is a product obtained after lysis of the cells grown during fermentation, is listed in the Cosmetics Ingredients Database (Cosing) as a skin conditioner.

Bioemulsifiers are compounds that contain biological molecules with surfactant properties similar to those of well-known synthetic surfactants [38]. Biosurfactants are amphiphilic molecules whose hydrophilic part consists of amino acids, peptides, esters, carbohydrates or hydroxyl phosphate, alcohol and carboxyl groups; the hydrophobic part consists of long-chain fatty acid residues, fatty acid β-hydroxyalkyls [39]. They are biodegradable, and this is one of their most important advantages because it prevents toxicity problems and accumulation in natural ecosystems [40]. Liposan was isolated by Cirigliano and Carman [41] from the nutrient medium supplemented with hexadecane by *C. lipolytica* ATCC 8662 (recently renamed to *Y. lipolytica*). Its composition was later determined, and studies have shown that it consists of 83% carbohydrate and 17% protein [42]. The composition of the bioemulsifier produced from the 1.5% (*w/v*) glucose-based fermentation of *Yarrowia lipolytica* was 47% protein, 45% carbohydrate, and 5% lipid [43]. The bioemulsifier, Yansan, was also isolated from a glucose-based fermentation medium containing 0.64% (*w/v*) glucose by *Y. lipolytica* (IMUFRJ 50682). Yansan is a lipid-carbohydrate-protein complex with high emulsification activity and stability in a wide pH range (3–9). Its composition was investigated and found to contain 62.1% carbon, 7.8% nitrogen, 29.2% oxygen, and 0.6% sulphur [40]. In a medium supplemented with 5% animal fat and 2.5% corn steep liquor, the EI index of the produced bioemulsifier against sunflower oil was 47% [44]. Da Silva et al. [39] fermented with 30 g/L crude residual glycerol, and the ferment liquids tested showed an average EI index of 56% after 24 h and 48 h.

The main hypothesis of this study was that an aerated *Yarrowia* broth has three phases (i.e., aqueous, solid, and foam phase), and each of them have important features which can establish the fundament of a *Y. divulgata*-based biorefinery. To verify this concept, we ran and developed erythritol fermentations and verified that its by-product foam phase is an effective bioemulsifier and that the by-product cell lysate is a useful cosmetic ingredient.

## 2. Materials and Methods

### 2.1. Fermentation

The strain was purchased from the National Collection of Agricultural and Industrial Microorganisms (Budapest, Hungary). The following microorganism was used in this study: *Yarrowia divulgata* (NCAIM 1485). The following media were used in this experiment:

Malt extract agar medium contains: Malt extract (Biolab, Budapest, Hungary) 30 g/L; Peptone (VWR, Leuven, Belgium) 5 g/L; Bacteriological agar (Reanal, Budapest, Hungary) 15 g/L.

Inoculum [45]: Glycerol (Carl Roth Gmbh+Co., Karlsruhe, Germany) 50 g/L; Yeast extract (Acros Scientific, Geel, Belgium) 3 g/L; Malt extract (Biolab, Budapest, Hungary) 3 g/L; Peptone (VWR, Leuven, Belgium) 5 g/L.

Fermentation medium [25]: Glycerol (Carl Roth Gmbh+Co., Karlsruhe, Germany) 100–150 g/L; Ammonium-chloride (Reanal, Budapest, Hungary) 4.56 g/L; $MgSO_4 \times 7\,H_2O$ (Donauchem, Budapest, Hungary) 1 g/L; Yeast extract (Acros Scientific, Geel, Belgium) 1 g/L; $CuSO_4$ $0.7 \times 10^{-3}$ (Reanal, Budapest, Hungary) g/L; $MnSO_4 \times H_2O$ $32.6 \times 10^{-3}$ (Reanal, Budapest, Hungary) g/L; $KH_2PO_4$ (Reanal, Budapest, Hungary) 97.8 g/L. In the experiment, 100 g/L glycerol was applied in all bioreactor experiments and Erlenmeyer's flasks, while 150 g/L was used for both normal Erlenmeyer and baffled flasks as well.

Inoculum cells for the fermenter and for the baffled flasks were grown in 250 mL Erlenmeyer's flasks containing 25 mL of inoculum media at 200 rpm at 25 °C on a rotary shaker for 72 h (New Brunswick Scientific (Edison, NJ, USA) Innova 40).

We used shaken baffled flask in triplicate and two sizes: 500 mL and 750 mL, with 50 mL and 75 mL working volume and 10% *v/v* inoculum with the same settings as the inoculum until the glycerol was exhausted from the culture media (ca. 8–10 days). The fermentations were carried out in a recently developed jFermi bioreactor with a Java-based web-client (https://jfermi.com/ (accessed on 31 July 2023)) and a working volume of 0.25 L, including 10% *v/v* inoculum. For production, the temperature was adjusted to 25 °C. For the bioreactor fermentations, the pH was decreased by the cells to pH = 3 and then controlled automatically to keep it at a level of 3 with addition of 6 N NaOH solution. Dissolved oxygen (DO) was controlled at 4 different levels (10–20–30–40%) by varying the stirrer rate with a PID controller. Standard deviations were determined through a duplicate run of DO = 40%, of which the relative error was used for estimating variance. If it was necessary, the foam was collected through a cyclone or reduced with polypropyleneglycol (PPG) antifoam in 1 mL/L.

Analytical Methods

Samples (1 mL) were taken from bioreactor and baffled flasks cultures; then, the samples were centrifuged for 5 min at 7500 rpm (Heraeus BIOFUGE pico, Hong Kong, China), and the cells pellets were washed with distilled water. The biomass was determined gravimetrically after drying at 105 °C (Sartorius MA35, Göttingen, Germany) and expressed in grams of cell dry weight per liter (g/L). Concentrations of glycerol, erythritol, and mannitol were measured from the supernatant by HPLC using a BioRadAminexHPX87H column at 65 °C, refractive index (Waters 2410 RI, Milford, MA, USA) detector at 40 °C, and Waters 1515 Isocratic pump. The column was eluated with 5 mM sulphuric acid with a flow rate of 0.5 mL/min, and samples were diluted 30 fold. Erythritol, mannitol, and glycerol standard were used.

For osmolarity measurement (Gonotec Osmomat 3000, Logan, UT, USA), 60 µL sample of the supernatant of the fermented media was used.

Sulphite oxidation method for $K_L a$ measuring:

In sulphite measurement, the measurement of the rate of oxygen absorption was reduced to the rate of a chemical reaction (1):

$$2SO_3^{2-} + O_2 \xrightarrow{Co^{2+}} 2SO_4^2 \tag{1}$$

The rate of the sulphation reaction was therefore determined only by the rate of oxygen absorption (2):

$$\frac{dC}{dt} = K_L a\left(C^* - C\right) = K_L a C^* \rightarrow r = K_L a C^* \tag{2}$$

where r—the oxidation rate; C*—saturation oxygen concentration ($mg/dm^3$); C—current dissolved oxygen concentration ($mg/dm^3$); $K_La$—oxygen absorption coefficient ($h^{-1}$).

The reaction rate was measured as follows: as the reaction time progressed, samples were taken in which the concentration of unreacted sulphite was determined; these values were plotted against time, and the directional tangent of the line was determined. The above equation assumes that the velocity is constant, so the concentrations decrease in a straight line with time. This slope was expressed in $mg\ SO_2/h^{-1}/1\ mL$ to give the sulphite number. Converting this to oxygen by stoichiometry gives the OTR ($mg\ O_2/h^{-1}/1\ mL$). By following the sulphite $\rightarrow$ sulphate conversion by iodometry, the OTR (oxygen transfer rate) can be calculated from the slope of the thiosulfate reduction/time plot of the reaction rate. The following Equation (3) was used for this:

$$\text{tg }\alpha = \frac{\Delta\left(\text{thiosulphate reduction} \times \text{thiosulphate factor}\left(cm^3\right)\right)}{\Delta\text{time}(min)} \tag{3}$$

Here, 1 mL of 0.1 n $I_2$ (or thiosulphate) solution is equivalent to 0.8 mg $O_2$. From the OTR = $K_La* C*$ obtained in this way, the $K_La$ value for the aeration/mixing conditions can be calculated [46]. The sulphite oxidation measurement was performed by titration, during which the loss of thiosulphate was monitored and plotted as a function of time. For the measurement at 25 °C, 0.1 n thiosulphate solution, 1 mL of 1 M hydrochloric acid solution, 10 mL of 0.1 n KI solution and 1–2 drops of 1% starch solution, and 1 mL of samples were used.

## 2.2. Skin Moisturising Measurement

For skin moisturising measurements, cells were separated at the end of fermentation. For cell separation, 10 mL of fermented broth was used, centrifuged (HERMLE Z 200 A, Wehingen, Germany) for 10 min at 6000 rpm speed, and then the supernatant was poured off and resuspended in 10 mL of distilled water. Cell disruption was performed by the IKA ULTRA TURRAX Tube Drive cell disruption system with glass beads in 3 mL volume of distilled water. Cell debris was removed by centrifugation at 9000 rpm for 1 min in distilled water. Short-term skin moisturising measurements were performed after treatments on $1 \times 1\ cm^2$ European human adult skin with the samples. The determination of skin moisture was carried out with the Multi Dermascope MDS 800 (Diepoldsau, Switzerland), as reported earlier by Tóth et al. [47], using the Corneometer of the equipment. After treatments, skin moisture content increased, decreased via the loss of excess humidity, and then finally stabilized. These final stabilized values were reduced with the untreated skin moisture values, and the differences were compared depending on different origin cell-lysates.

## 2.3. Emulsifying Activity Measurement

This experiment was based on the measurement of Czinkóczky et al. [48]. For the measurement, a supernatant of fermented broth was used, of which 2 mL was added to 2 mL of sunflower oil in a test tube and vortexed for 2 min. The following Equation (4) was used for the evaluation:

$$EI_t = (H_e/H_t) \times 100 \tag{4}$$

where $H_e$ and $H_t$ are the height of emulsion and total height of the liquid in the tube, respectively. The tubes were incubated at 25 °C for 1 day. The emulsification index (EI, %) was determined after 1 h, and the EI was measured after 24 h (EI24, %).

## 2.4. Statistical Analysis

Where indicated, repeated experimental setups were averaged and compared with a 2-sample *t*-test for detecting significancy with MS Excel. For evaluating the effect of $K_La$

and glycerol on erythritol concentration, we used SigmaPlot 7 (SPSS Inc., Chicago, IL, USA) for non-linear (3D) curve fitting, which resulted in Equation (5) with $R^2 = 0.88$.

$$\text{erythritol} = \frac{776.233}{\left(\frac{K_La - 1425.1172}{201.3215}\right)^2 \times \left(1 + \left(\frac{\text{Glycerol} - 99.7236}{116.4087}\right)^2\right)} \tag{5}$$

## 3. Results

### 3.1. Fermentations

#### 3.1.1. Fermenter

The effect of oxygen on erythritol fermentation was investigated in the jFermi fermenter. Different oxygen levels (10%; 20%; 30%; 40%) were controlled by automatically varying the impeller speed. One of the best results of such fermentations with DO = 20% oxygen level is shown in Figure 1.

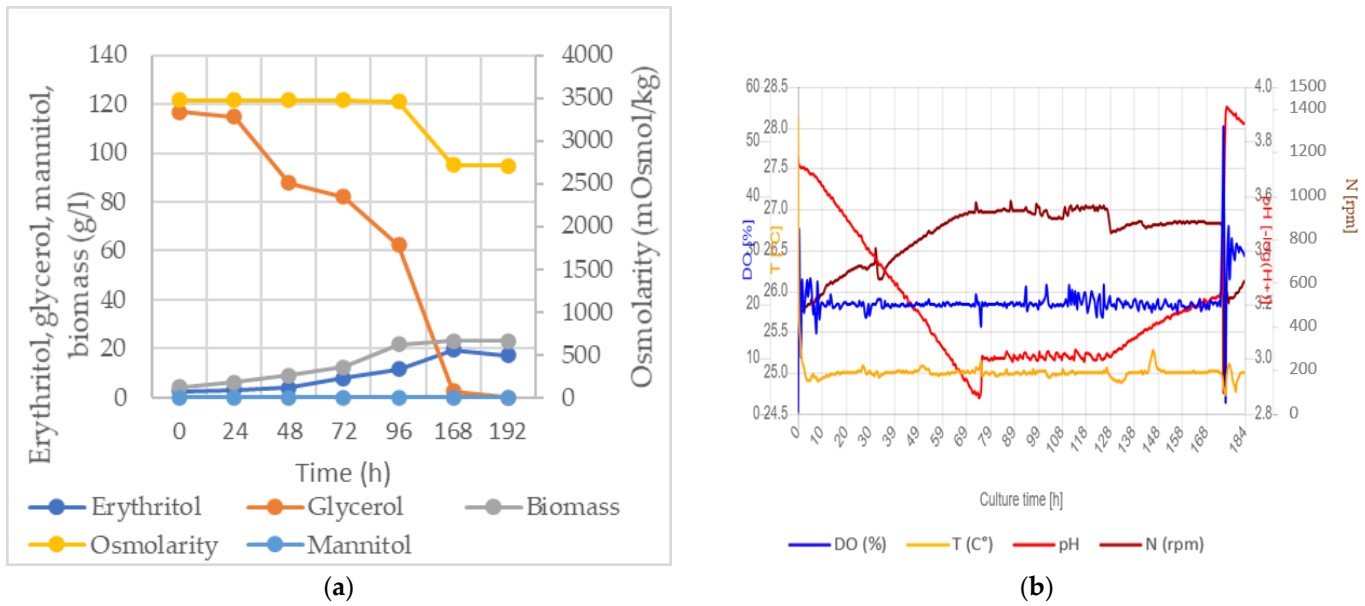

**Figure 1.** Fermentation at DO = 20% oxygen level. (**a**) Parameters measured off-line during fermentation. (**b**) Online measured and controlled parameters.

The fermentation produced 19.65 g/L erythritol, with no mannitol as a by-product. The osmolality was decreased from an initial 3474 mOsmol/kg to 2712 mOsmol/kg. The glycerol completely consumed from the ferment media. Figure 1b clearly shows that the pH was decreased to under 3 and controlled with 6 N NaOH to 3. The dissolved oxygen level was kept at 20% during the fermentation. The biomass at the end of the fermentation was 23.1 g/L. The rise in pH was marked the end of fermentation, after which it was difficult to keep the oxygen level at 20%; later, it also rose. The results of the fermentations at other oxygen levels are shown in Table 2.

Most erythritol was obtained during the batch fermentation with 40% oxygen, $19.81 \pm 3.47$ g/L, but a lot of glycerol (41.98 g/L) remained in the ferment media. Furthermore, the highest yield of $28.85 \pm 5.83\%$ was obtained here. However, the productivity was $0.08 \pm 0.053$ (g/L)/h lower than the others due to the 10 days of fermentation. The overall polyol production was the highest at the 30% oxygen level, where 17.75 g/L erythritol and 13.77 g/L mannitol were produced as by-products. At the 20% oxygen level, 19.65 g/L erythritol was produced, and at this fermentation setting, glycerol was completely consumed from the culture medium.

**Table 2.** Results of the *Y. divulgata* fermentations under different oxygen levels.

| Oxygen Level (%) | Eryhtritol (g/L) | Mannitol (g/L) | Initial Osmolarity (mOsmol/kg) | Residual Glycerol (g/L) | Biomass (g/L) | $Y_{Ery}$ (%) | Productivity (g/L)/h |
|---|---|---|---|---|---|---|---|
| 10 | 16.66 ± 2.92 | 0 | 3405 ± 57.88 | 23.8 ± 3.84 | 20.9 ± 0.94 | 22.99 ± 4.64 | 0.06 ± 0.039 |
| 20 | 19.65 ± 3.44 | 0 | 3474 ± 59.05 | 2.72 ± 0.43 | 23.1 ± 1.04 | 14.8 ± 2.99 | 0.12 ± 0.079 |
| 30 | 17.75 ± 3.1 | 13.77 ± 2.73 | 3475 ± 59.07 | 13.33 ± 2.15 | 23.62 ± 1.06 | 14.29 ± 2.88 | 0.123 ± 0.08 |
| 40 | 19.81 ± 3.47 | 2.62 ± 0.52 | 3439 ± 58.46 | 41.98 ± 6.78 | 21 ± 0.94 | 28.85 ± 5.83 | 0.08 ± 0.053 |

$Y_{Ery}$ erythritol yield.

### 3.1.2. Comparison of Oxygen Absorption

The above results suggested that DO has a special role in erythritol fermentation in accordance with previous reports [28,29]. Moreover, in some of our preliminary shaking flask experiments, we detected even higher product formation. Therefore, we compared the oxygen absorption ($K_La$) between the two types of flasks and the jFermi bioreactor. The jFermi $K_La$ was determined by the manufacturers using the "Gassing out" method, with Q = 200 mL/min at 300 and 500 rpm resulting in values of 9.56 and 16.72 $h^{-1}$, respectively. Based on this, the $K_La$ value of the fermenter with a varied stirring rate between 600 and 1000 rpm, thus considered as an average speed of 800 rpm, was $K_La$ = 26.4 $h^{-1}$.

The oxygen absorption coefficient ($K_La$) of the two type flasks was determined and compared by the sulphite oxidation method at a 3-3 different shaking (stirring) rate in 500 mL flasks; the results are shown in Figure 2. The slope of the lines show that the baffled flask had a $K_La$ that was 2.45-fold higher than the classic Erlenmeyer flask of same size and filling. When a typical stirring rate (200 rpm) was used for $K_La$ calculation, a $K_La$ of 267.12 $h^{-1}$ and 655.74 $h^{-1}$ were observed for normal and baffled flasks, respectively. This highlights that these shaking flask setups had ca. 10–25 fold-higher oxygen absorption compared to the jFermi bioreactor.

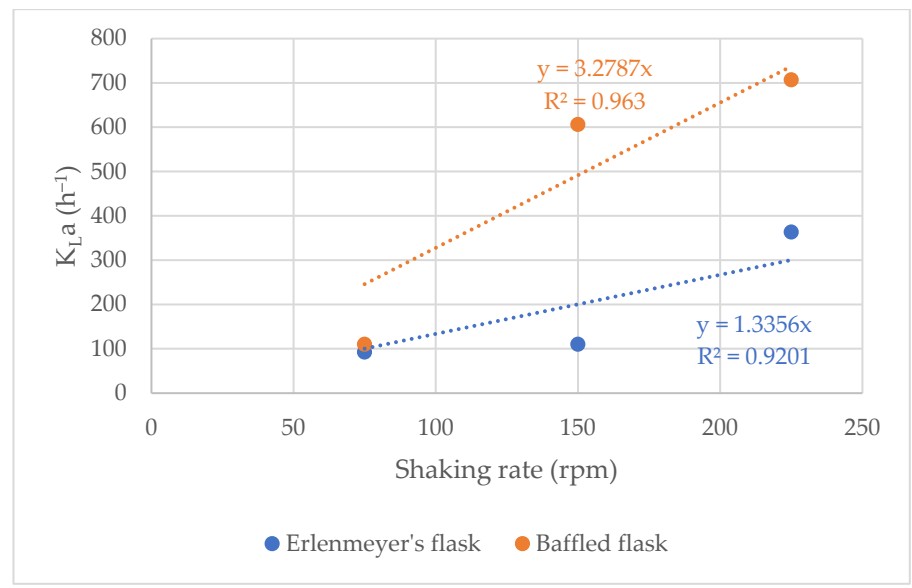

**Figure 2.** Comparison of normal flask and baffled flask $K_La$ measurements.

### 3.1.3. Baffled Flasks Experiments

For higher erythritol production, shaking flasks were used with an increased initial glycerol concentration (150 g/L), thus also increasing the osmolarity during fermentation. During the fermentation in the normal flask, 14.5 g/L erythritol was produced, yielding 34.5% in 10 days. The glycerol was not well utilized by the strain during the fermentation,

leaving 105.7 g/L in the fermented media on day 10. To achieve higher erythritol yields, the effect of increased aeration on erythritol production was investigated in baffled flasks. Table 3 summarises the results of the fermentations in 500 mL and 750 mL baffled flasks. Whereas yields varied during fermentations the highest yields of each experiments are indicated in the Table 3.

**Table 3.** Results of fermentations in the Erlenmeyer's flask and baffled flasks.

| Fermentations | Erythritol (g/L) | Yield (%) | Productivity (g/L)/h | Biomass (g/L) |
|---|---|---|---|---|
| Erlenmeyer's flask 500 mL | $14.5 \pm 1.1$ | $34.5 \pm 7.2$ | $0.06 \pm 0.006$ | $6.21 \pm 0.28$ |
| Baffled flask 500 mL | $44.14 \pm 1$ | $26.43 \pm 3.37$ | $0.19 \pm 0.03$ | $27.83 \pm 0.24$ |
| Baffled flask 750 mL | $42.42 \pm 5.08$ | $27.74 \pm 7.71$ | $0.19 \pm 0.02$ | $23.53 \pm 2.1$ |

The 500 mL baffled flask produced an average of $44.14 \pm 1$ g/L of erythritol over the three replicates, while the 750 mL baffled flask produced $42.42 \pm 5.08$ g/L. The average yield of fermentations in the three replicates was $26.43 \pm 3.37\%$ for the 500 mL flasks and $27.74 \pm 7.71\%$ for the 750 mL flasks. The $p$-value of the paired $t$-test was 0.842132, so there was no significant difference in the yield of fermentations in different flask sizes. Productivity averaged $0.19 \pm 0.03$ (g/L)/h for fermentations in 500 mL flasks and $0.19 \pm 0.02$ (g/L)/h for 750 mL flasks.

### 3.2. Skin Moisture Effect Determination

Since *Y. lipolytica* cell lysate is registered cosmetic ingredients in the EU, we studied the same effects of *Y. divulgata* lysates.

The highest skin moisturizing effect was found at the 20% oxygen level, where the initial skin moisture was increased by 12.58%, which is a significant ($p$ = 0.048874) change at the 0.05 significance level and is detailed in Table 4.

**Table 4.** Calculated moisturising effect of the *Yarrowia* ferment lysate with the measured results in the fermentor.

| Oxygen Level (%) | Mean of Initial Values (%) | Mean of Final Measured Values (%) | Calculated Moisturising Effect (%) | *p* * |
|---|---|---|---|---|
| 10 | $59.67 \pm 1.15$ | $59.33 \pm 2.31$ | $-0.56 \pm 3.06$ | 0.704833 |
| 20 | $53 \pm 0$ | $59.66 \pm 0.58$ | $12.58 \pm 0.58$ | 0.048874 |
| 30 | $47 \pm 1$ | $49.66 \pm 0.58$ | $5.67 \pm 1.53$ | 0.267720 |
| 40 | $49.66 \pm 1.52$ | $50.66 \pm 2.08$ | $2.01 \pm 1$ | 0.640983 |

* $p < 0.05$ was considered indicative of significance ($t$-test).

Cell lysate, which was fermented at 10%, did not cause any significant difference from the original skin moisturizing effect. Cell lysates grown under 30% and 40% oxygen levels were almost able to increase skin hydration, the former by 5.67% and the latter by 2.01%.

Cell disruption was also performed after fermentations in baffled flasks, after which we measured skin hydration. In each case, the measurement was carried out from the last sample, in which the erythritol level was no longer at its maximum level.

Table 5 shows the hydration results of the cell lysate from flask fermentation, including the initial and final values and which had a significant effect. In all cases, the tested lysates increased skin hydration by $9.11 \pm 5.47\%$ for the 500 mL flask and by $11.86 \pm 6.25\%$ for the 750 mL flask. No significant difference was observed between the two differently sized baffled flasks ($p$ = 0.613744).

**Table 5.** Calculated moisturising effect of the *Yarrowia* cell lysate with the measured results in the baffled flasks.

| Fermentations (500 mL) | Mean of Initial Values (%) | Mean of Initial Measured Values (%) | Mean of Final Values (%) | Mean of Final Measured Values (%) | Calculated Moisturising Effect (%) | Mean of Calculated Moisturising Effect (%) | *p* * |
|---|---|---|---|---|---|---|---|
| 1. | 41.66 ± 0.57 | | 48 ± 2.64 | | 15.2 ± 2.3 | | |
| 2. | 53 ± 1.73 | 47.1 ± 5.68 | 57 ± 2.64 | 51.11 ± 5.1 | 7.54 ± 3.6 | 9.11 ± 5.47 | 0.675562 |
| 3. | 46.66 ± 7.25 | | 48.33 ± 6.43 | | 4.59 ± 3.38 | | |
| **Fermentations (750 mL)** | **Mean of Initial Values (%)** | **Mean of Initial Measured Values (%)** | **Mean of Final Measured Values (%)** | **Mean of Final Measured Values (%)** | **Calculated Moisturising Effect (%)** | **Mean of Calculated Moisturising Effect (%)** | ***p* *** |
| 1. | 36.6 ± 0.57 | | 42.6 ± 2.08 | | 16.36 ± 2 | | |
| 2. | 43.66 ± 0.57 | 43.19 ± 6.37 | 50 ± 1.73 | 48.08 ± 4.82 | 14.5 ± 1.15 | 11.86 ± 6.25 | 0.380797 |
| 3. | 49.33 ± 0.57 | | 51.66 ± 1.15 | | 4.72 ± 0.57 | | |

* $p < 0.05$ was considered indicative of significance (*t*-test).

### 3.3. Emulsifying Activity Measurements

Several studies have shown the ability of *Yarrowia lipolytica* to produce bioemulsifiers; therefore, the emulsification index of the supernatant was measured at the end of all *Y. divulgata* fermentations. The emulsification index determines the ability of a molecule to emulsify hydrocarbons, and to be considered an effective emulsifier, the molecule must maintain at least 50% emulsification after 24 h of rest [39].

The highest emulsification index was obtained after 1 h (92.07%) in fermentation under 20% oxygen, as shown in Figure 3. The EI24 index of the same sample decreased to 55.06% but remained in the effective emulsification range. For all oxygen levels, EI24 remained above 50%, making it an effective bioemulsifier.

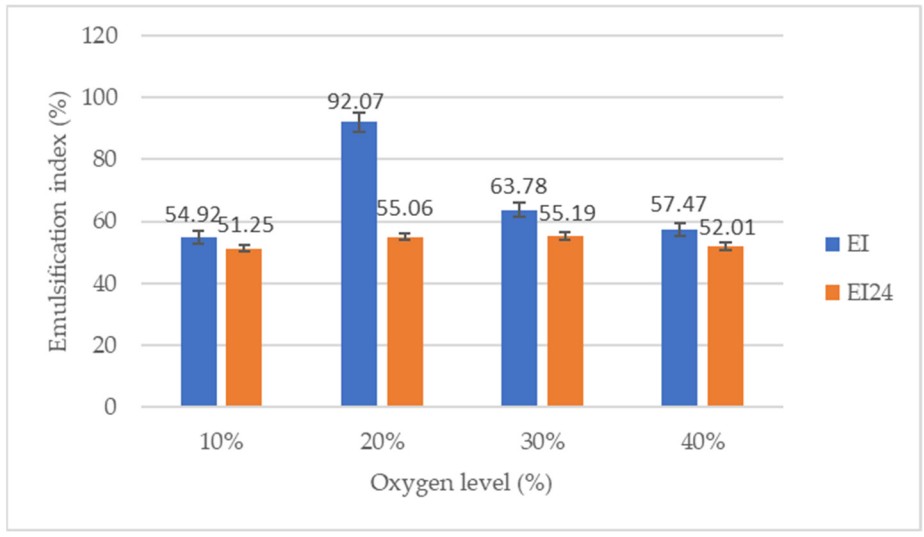

**Figure 3.** Emulsification index (%) results at different oxygen levels in the jFermi bioreactor.

The emulsification index measurement was also performed for the baffled flasks; the results are summarised in Table 6, with an EI24 of 53.63 ± 1.85% for the 500 mL flasks and 55.58 ± 6.76% for the 750 mL flasks. Since all remained above 50% after 24 h, they can all be considered as stable bioemulsifiers.

**Table 6.** Emulsification index (%) results in baffled flasks.

| Fermentations (500 mL) | Emulsification Index 1 h (%) | Mean of Emulsification Index 1 h (%) | Emulsification Index 24 h (%) | Mean of Emulsification Index 24 h (%) |
|---|---|---|---|---|
| 1. | 82.6 | | 52.15 | |
| 2. | 67.38 | 68.55 ± 13.5 | 54.91 | 53.63 ± 1.85 |
| 3. | 55.67 ± 1.96 | | 53.85 ± 1.17 | |
| **Fermentations (750 mL)** | **Emulsification Index 1 h (%)** | **Mean of Emulsification Index 1 h (%)** | **Emulsification Index 24 h (%)** | **Mean of Emulsification Index 24 h (%)** |
| 1. | 64.36 | | 63.40 | |
| 2. | 52.5 | 56.45 ± 6.83 | 51.7 | 55.58 ± 6.76 |
| 3. | 52.5 | | 51.64 | |

## 4. Discussion

The results were showed that for efficient erythritol fermentation (44 g/L erythritol with 0.19 g/L/h productivity), a $K_La$ range of 250–660 $h^{-1}$ was needed beside a minimal osmolarity of 2500 mOsmol/kg. As our bioreactor experiments show, maintaining a constant DO level may decrease the erythritol concentration above 20–40%. This may be caused by the inhibition of the erythrose reductase enzyme by the high oxygen level. However, the high oxygen transfer rate produced by higher $K_La$ is beneficial since it enhances biomass formation, which also increases the primary metabolite product formation. The effect of different glycerol concentrations and $K_La$ values on erythritol product concentration is plotted on Figure 4. It indicates that the amount of glycerol had lower impact than $K_La$, verifying the importance of the oxygen transfer rate for efficient erythritol production.

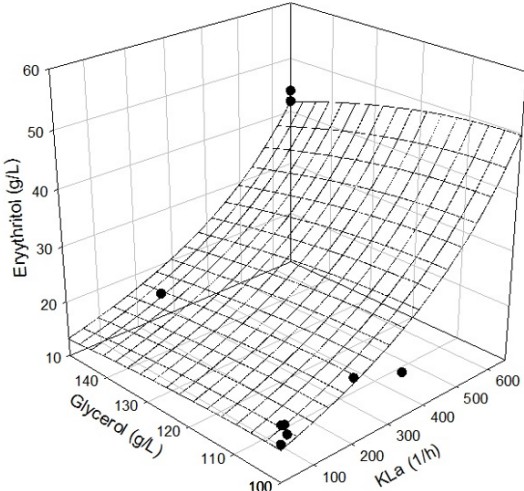

**Figure 4.** Erythritol dependency on glycerol concentration and $K_La$.

However, the skin hydrating results show that glycerol had a more intensive effect on skin hydration compared to $K_La$. This might be because *Y. divulgata* grown on glycerol forms intracellular oil containing oleic acid and linoleic acid [49], which are effective skin emollient-forming films on the skin. Thus, higher oil content may better prevent skin dehydration, i.e., may enhance skin moisturizing. The higher glycerol amount increased the C/N ratio, which enhanced the secondary metabolite, i.e., oil formation. Magdouli et al. also studied the effect of aeration on *Yarrowia* oil production and found an optimum level below 30% [49]. This is also in good agreement with our skin moisturizing results, where highest value was observed at DO = 20%. The bioemulsifier (EI24) results correlate with

the skin moisturizing effect (despite the fact that the previous one is extracellular and the latter is intracellular), most probably because it is also a secondary metabolite.

## 5. Conclusions

Our study focused on a multi-purpose utilization of *Y. divulgata* which can promote its application in a microbial biorefinery. The main product was erythritol, a good sugar-replacing alternative. Regarding its production, maintaining a constant DO level above 20% was not beneficial because a high oxygen level inhibited the erythritol formation via erythrose reduction. The separated *Y. divulgata* cell's lysate was found to be an effective skin moisturizer, which could increase skin hydration by more than 10% due to its oil content for which higher glycerol and higher $K_La$ values seemed to beneficial. Similar results were observed for the bioemulsifying effect of *Y. divulgata* fermentation broth. Since a renewable carbon source (glycerol) was used, and the resulting fermentation broth as well as separated cells can be used in high-value products, *Y. divulgata* is a good candidate for use in biorefineries.

**Author Contributions:** Conceptualization, E.E. and Á.N.; methodology, Á.N.; investigation, E.E.; data curation, E.E.; writing—original draft preparation, E.E.; writing—review and editing, Á.N.; visualization, E.E.; supervision, Á.N. All authors have read and agreed to the published version of the manuscript.

**Funding:** The APC was funded by MÉL Biotech Kft (Budapest, Hungary).

**Informed Consent Statement:** Not applicable.

**Data Availability Statement:** The data presented in this study are available on request from the corresponding author.

**Acknowledgments:** The authors are very grateful to MÉL Biotech K+F Kft (AFo Biotech R&D Ltd.) for providing Ultra Turax Tube and MDS800 Multidermascope equipments.

**Conflicts of Interest:** The authors declare no conflict of interest.

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
