# Peer review of "Investigations for a Yarrowia-Based Biorefinery: In Vitro Proof-of-Concept for Manufacturing Sweetener, Cosmetic Ingredient, and Bioemulsifier"

_fermentation, doi:10.3390/fermentation9090793_

Round 1

Reviewer 1 Report

The present work is an interesting study on Y. divulgata, tested from different points of view. However it looks chaotic and it is not always easy to understand what has been done. A personal impression: it seems that in the same work have been incorporated many different aspects,   which, instead, if treated individually and in more depth, would have led to more publications. For example the authors could have focused on oxygen, exploring other stressors, such as nitrogen content, that can impact Y metabolism. However, this is only a personal suggestion

Abstract

The abstract looks like a good summary of the work, with a short but comprehensive presentation. However, more attention to English writing is recommended.

Introduction

The introduction is clearly presented and consists of a comprehensive literature review. Some parts could be improved

Lines 33-43: Authors could pay more attention to the English and make reading more fluid by arguing each sentence more (at the moment this part of the introduction looks like a list).

Line 37: as welL as

Line 47: “indistinguishable from Y. lipolytica and Y. deformans” from what point of view? Some characteristics must allow them to be distinguished

Lines 56-58: Low calorie intake is not the reason. Quite simply, consuming erythritol does not interfere with blood sugar. Please correct the sentence

Line 75: representS

Line 126-128: perhaps the sentence could be rephrased as follows: The Yarrowia lipolytica fermentation lysate, which is a product obtained after lysis of the cells grown during fermentation, is listed in the Cosmetics Ingredients Database (Cosing) as a skin conditioner.

Material and Methods

The section is chaotic in some parts and poor of some details. For all reagents, the manufacturing company and address must be indicated. There is no statistical analysis

Lines 162-165: the sentence could be reformulated: “Both in the experiments with the fermenter and in the one with the Erlenmeyerf lask the medium consisted of …..The only difference was the glycerol which in the experiments with the fermenter was 100 g/L, while in that with the Erlenmeyer flasks was 150 g/L”. The authors could also explain the reason for the different concentrations. Also, didn't the authors perform any experiments with the same concentration of glycerol between the fermenter and the flasks?

Lines 166-168: How long does the inoculum preparation/revitalization take?

Lines 171-172: How long does the fermentation take?

Lines 181-182: please give more details about centrifugation

Line 222: skin moisture conTent

Lines 222-226: the methods are not very clear

Results

Result section is quite cahotic and should be improved

Lines 240-241: The best results of the fermentations were reached with D=20%, and the main results are shown in Figure 1. The results obtained with all DO values are instead summarized in table 1

Line 244: please delete “was”

Line 246: do the authors mean that the glycerol was completely consumed in the fermentation medium?

Line 246: please delete “IN” before Figure

Figure 1b: a legend is needed, because the figure as it stands is very difficult to interpret.

Table 4: if I understand correctly, only DO=20% determines a statistically significant result. Therefore, there is no difference between 10, 30 and 40%. It's correct? If so, lines 307 308 should be rewritten

Figure 4: it seems quite redundant

Discussion

The discussion seems a repetition of the results and there isn’t comparison with the literature. I think this greatly weakens the work. Probably gradually discussing the results, arguing them with comparisons, would greatly improve the work.

Author Response

It's the Authors' pleasure to receive Your remarks and comments, thank You for Your efforts in the improvement of our manuscript! Please see the detailed answers in the attached file.

Reviewer 2 Report

I am very grateful you for the invitation to review manuscript fermentation-2562769 by Eszterbauer and coauthors "Investigations for a Yarrowia based biorefinery: in vitro proof of concept for manufacturing sweetener, cosmetic ingredient, and bioemulsifier”. The aim of this work was to investigate Y. divulgata’s complex utilisation based on erythritol fermentation to establish a Yarrowia based biorefinery in which both the fermentation broth and separated cells are utilised in high-value products. The work is interesting but needs several adjustments to increase the quality of the material.

Comments:

- Abstract, line 14: Include products commonly produced by Y. lipolytica.

- Abstract, line 18: Specify high value-added products.

- Abstract: Please indicate a better step-by-step about the work, including conditions used.

- Line 21: Change “The results were showed that” to “The results showed that”.

- Line 22: Change “g/l” to “g/L”. Check standardization throughout the entire text.

- Line 22-23: It is not clear how it was evaluated for this claim.

- Lines 25-28: The data presented in the abstract do not allow the affirmation of the conclusion.

- Lines 29-30, Keywords: Change the repeated keywords by different words from the title.

- Introduction: Complement and update data from the introduction with more recent works (https://doi.org/10.3390/pr10020381 and 10.3390/molecules27072300 for example).

- Line 42-43: Include other components in addition to erythritol.

- Line 48: Chang “relative.. A few” to “relative. A few”.

- Introduction: Please include data on the market and usage of erythritol.

- Line 69-71: Note the use of italic style.

- Introduction: Information about the biorefinery should be added since it is the focus of the work.

-  Line 175: Specify the DO term in the first presentation (although common).

- Line 209: Change “calculated. [39].” to “calculated [39].”.

- Line 229: Review the use of the term “fermented juice” in this context or further explain the methodology.

- Line 242: Figure 1b. Specify the reference of the lines according to the color, as shown in the caption of Figure 1a.

- Table 2: Specify which component the Yield term is related to.

- Lines 265, 269 and throughout the text: Standardize the “Kla” presentattion.

- Line 287: Change “Table 3. summarises” to “Table 3 summarises”.

- Figure 3 and Table 4 present the same results. Remove Figure.

- Line 314: Change “Figure 4. shows” to “Figure 4 shows”. Review this error throughout the text.

- Discussion, Lines 345-349: This is repetition of objectives and material and methods. No new information is entered.

-  Line 350: Why? Discussion is aimed at theoretical explanations.

- Discussion: No discussion is presented in this item. The authors only repeat results already presented.

- Discussion: the discussion should be used to promote the discussion of biochemical parameters, operational differences, among others.

- No discussion about “biorefinery”, process viability, among other important parameters was presented.

- Conclusion: No conclusion.

- Authors must include an in-depth discussion on the topic.

Author Response

(The authors gave the same response as above.)

Round 2

Reviewer 1 Report

Thanks to the authors for replying and editing the manuscript. A small revision should be made (pay attention to the numbering of the references)

Author Response

Thank You for Your kind remark, we renumbered the references so now they are continuously increasingly cited starting from [1] up to [49]. Thanks again for all of Your help!

Reviewer 2 Report

After carefully checking the responses and the revisions, the manuscript is suitable for Fermentation.

Author Response

Thank You for Your kind revision and support of enhancing our manuscript!